# Quasiadiabatic electron transport in room temperature nanoelectronic devices induced by hot-phonon bottleneck

Qianchun Weng [1,2,10,11✉], Le Yang[3,11], Zhenghua An [3,4,11✉], Pingping Chen[1], Alexander Tzalenchuk [5,6], Wei Lu[1,7✉] & Susumu Komiyama[1,8,9]

Since the invention of transistors, the flow of electrons has become controllable in solid-state electronics. The flow of energy, however, remains elusive, and energy is readily dissipated to lattice via electron-phonon interactions. Hence, minimizing the energy dissipation has long been sought by eliminating phonon-emission process. Here, we report a different scenario for facilitating energy transmission at room temperature that electrons exert diffusive but quasiadiabatic transport, free from substantial energy loss. Direct nanothermometric mapping of electrons and lattice in current-carrying GaAs/AlGaAs devices exhibit remarkable discrepancies, indicating unexpected thermal isolation between the two subsystems. This surprising effect arises from the overpopulated hot longitudinal-optical (LO) phonons generated through frequent emission by hot electrons, which induce equally frequent LO-phonon reabsorption ("hot-phonon bottleneck") cancelling the net energy loss. Our work sheds light on energy manipulation in nanoelectronics and power-electronics and provides important hints to energy-harvesting in optoelectronics (such as hot-carrier solar-cells).

[1] National Laboratory for Infrared Physics, Shanghai Institute of Technical Physics, The Chinese Academy of Sciences, Shanghai, China. [2] Institute of Industrial Science, The University of Tokyo, Meguro-ku, Tokyo, Japan. [3] State Key Laboratory of Surface Physics, Institute for Nanoelectronic Devices and Quantum Computing, and Department of Physics, Fudan University, Shanghai, China. [4] Shanghai Qi Zhi Institute, Xuhui District, Shanghai, China. [5] National Physical Laboratory, Teddington, UK. [6] Royal Holloway, University of London, Egham, UK. [7] School of Physical Science and Technology, ShanghaiTech University, Shanghai, China. [8] Department of Basic Science, The University of Tokyo, Meguro-ku, Tokyo, Japan. [9] Terahertz Technology Research Center, National Institute of Information and Communications Technology, Koganei, Tokyo, Japan. [10] Present address: Surface and Interface Science Laboratory, RIKEN, Wako, Saitama, Japan. [11] These authors contributed equally: Qianchun Weng, Le Yang, Zhenghua An. ✉email: anzhenghua@fudan.edu.cn; qcweng@gmail.com; luwei@mail.sitp.ac.cn

In modern semiconductor electronic devices, current-carrying electrons are locally driven far away from equilibrium (with an effective electron temperature $T_e$ largely exceeding the lattice temperature $T_L$)[1,2], and these hot electrons accelerate/decelerate frequently to fulfil intended functions. The excess energy of hot electrons typically dissipates locally to the lattice as the Joule heat, which not only leads to a major heat concern for post-Moore-era nanoelectronics[3] but also exert a thermodynamic limitation to the energy efficiencies in all solid-sate electronic devices (such as the Shockley–Queisser limit[4], being only ~30% for Si solar cells). To suppress net energy loss to lattice, the excess energy carried by hot electrons has to be transmitted along with the charge transport. Nevertheless, dissipationless charge transport is hitherto realized only in, e.g., superconductors[5,6] or topological transistors[7] under (quasi-)equilibrium limit ($T_e T_L$) or nanoscale vacuum transistors[8] where phonon-emission process is eliminated. For electrons in the strong nonequilibrium conditions ($T_e \gg T_L$), however, dissipationless transport appears to be challenging and has never been addressed because electron cooling occurs spontaneously at an intrinsically fast speed due to rapid electron–phonon interactions (~ps).

Numerous works have been devoted to exploiting the exotic properties of the hot electrons in strong nonequilibrium conditions such as superdiffusion[9], chaotic diffusion[10], thermal oscillation[11], and prototype devices have been attempted such as hot-electron transistors[12], hot luminescent light sources[13], highly efficient solar cells[14] and plasmon-enhanced photochemistry[15,16], etc. In all these works, however, the transport of the hot electrons remains to be highly dissipative which restricts severely the achievable device performance. Further improving the device performance requires a comprehensive understanding of nanoscale kinetics of how exactly the energy is carried by the nonequilibrium electrons, and how the energy dissipation to the lattice can be significantly suppressed. This, however, eluded direct nanothermometric observation in real space due to nonequilibrium nature of embedded electrons and their intrinsically small heat capacity (typically several orders of magnitude less than that of the lattice). A number of highly sensitive nanoscale mobile carrier[17] or current imaging[18] and scanning nanothermeometry techniques[19–22] have recently been developed, revealing critical local information about charge transport and heat generation, but they are insensitive to the electron temperatures and do not necessarily provides straightforward access to the detailed interplay between the strongly nonequilibrium electrons and their host lattice systems.

Here, by using recently developed radiative electronic nanothermometry[23] together with a conventional contact-type one, we separately imaged and compared heated electrons ($T_e$) and lattice ($T_L$) in GaAs/AlGaAs quantum well (QW) conducting channels. With these real-space measurements, we disclosed diffusive, but nearly dissipationless transport of hot electrons at room temperature: More than 90% of the electron energy passes unexpectedly through a channel of up to 1 μm length without substantial dissipation to the lattice (much longer than the mean free path). The dramatic suppression of the energy loss across this ultralong distance is attributed to overpopulated hot longitudinal–optical (LO) phonons that induce frequent LO-phonon reabsorption and thereby remarkably slow down the electron cooling. This hot-phonon-assisted electron transport is reminiscent of the previously reported "hot-phonon bottleneck effect" for photoexcited transient carriers (e.g., in perovskites)[24–29]. Noting that the hot-phonon bottleneck effect is exempted from any restriction of device operating temperatures, our observations may find promising applications in on-chip energy management for solid-state electronics and energy-harvesting technologies.

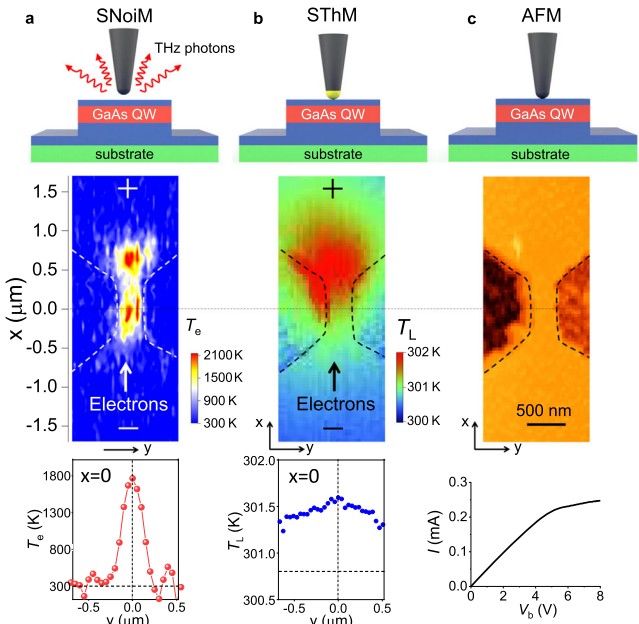

**Fig. 1 Nanoscale thermal probing of heated electrons and heated lattice. a** SNoiM. Top panel: A non-contact probe tip, 10 nm above the sample surface, scatters fluctuating THz evanescent fields for detection. Middle panel: A 2D image of $T_e$ on a narrow conducting channel biased with $V_b = 8$ V. Bottom panel: A 1D profile of $T_e$ across the channel at $x = 0$. The signal averaging time is 300 ms/pixel for each pixel = 50 nm × 50 nm, where low-frequency fluctuation in the signal roughly corresponds to $\Delta T_e$ ~±100 K in the region outside mesa (without conduction electrons) and $\Delta T_e/T_e = \pm7\%$ in the region of mesa (with conduction electrons). **b** SThM. Top panel: A nanothermometer (shown with yellow dots) is in thermal contact with the sample surface to measure $T_L$. Middle panel: A 2D image of $T_L$ taken on the same device as that for SNoiM with $V_b = 8$ V. Bottom panel: A 1D profile of $T_L$ across the channel at $x = 0$. The dotted horizontal line shows the base temperature ($T_L = T_{base} = 300.8$ K). The signal averaging time is ~ 8 ms/pixel for each pixel = 40 nm × 40 nm, where relative precision of measurements is roughly $\Delta T_L$ ~±100 mK. **c** AFM. Top panel: AFM tip. Middle panel: A 2D topographic image of the device studied, with a 400 nm wide and 870 nm long constriction channel of a quasi 2D electron system in a GaAs/AlGaAs quantum well (see "Methods"). Bottom panel: current–voltage trend.

## Results and discussion

**Nanoscale thermometric imaging of both conduction electrons and the lattice.** The non-contact electronic nanothermometry (top panel of Fig. 1a) is a scanning noise microscope (SNoiM) that has recently proven to sensitively detect the shot noise generated by hot electrons[23,30]: hot-electron distribution in real space is thereby visualized noninvasively. (see "Methods" and Supplementary Note 1). In this technique, a sharp metal tip scatters fluctuating electromagnetic evanescent fields at terahertz (THz) frequencies (about 20.7 ± 1.2 THz) that are generated on a sample surface by current fluctuation of conduction electrons, and the scattered THz wave is collected and detected by an ultrahigh sensitivity detector called a charge sensitive infrared phototransistor[31]. The second method is contact-type scanning thermal microscopy (SThM)[19], where a thermistor is integrated with an atomic force microscope cantilever, which locally probes the lattice temperature (top panel of Fig. 1b) via contact thermal conduction rather than conduction electrons (see "Methods" and Supplementary Note 2).

All the measurements are made at ambient temperature ($T_{Room} \approx 300$ K). The middle panel of Fig. 1c shows the atomic

force microscopy (AFM) topography of a narrow GaAs/AlGaAs QW heterostructure device with the current–voltage characteristics displayed underneath (see "Methods"). The middle panel of Fig. 1a displays a representative two-dimensional (2D) colour plot of the effective electron temperature $T_e$, converted from the SNoiM signal (see "Methods") at bias voltage $V_b = 8.0$ V (and the corresponding source-drain current is $I_{ds} = 0.24$ mA). White broken lines are guides for the eyes and mark the edge that defines the constriction channel. The SNoiM signal exists only in the channel region, as elucidated explicitly in the bottom panel of Fig. 1a, with a one-dimensional (1D) profile of $T_e$ taken across the constriction channel (in the $y$-direction at $x = 0$). This indicates that electrons are probed and not the lattice. Notably, two distinct hot spots are recognized, one close to the entrance for electrons in the constricted channel and the other outside the channel close to the exit. The highest temperatures at the hot spots reach $\Delta T_e \sim$ 1700 K for $T_e = T_{Room} + \Delta T_e \approx 2000$ K.

The middle panel of Fig. 1b displays a 2D colour plot of the SThM signal and shows a profile of $T_L$. In contrast to that of $T_e$, the profile is featured by a broader structure with a single peak located outside the exit. No structure is discerned corresponding to the electron hot spot on the entrance side. In addition, the heated region is not confined in the conducting channel but smoothly spreads out of the channel, as explicitly shown in the bottom panel of Fig. 1b, with 1D profile across the channel (in the $y$-direction at $x = 0$) (see also Supplementary Fig. 4) The distribution of $T_L$ is smooth and spread. There is a small temperature rise such that $\Delta T_L \sim 1$ K at the maximum, where $T_L = T_{base} + \Delta T_L$ and $T_{base} = 300.8$ K (see "Methods" and Supplementary Fig. 3 and Supplementary Note 3), which is due to a large lattice specific heat and the fact that the heat spreads via lattice thermal conduction[32].

The double-peak structure of $T_e$, studied with SNoiM, contrasted with the single-peak structure of $T_L$, is shown in Fig. 2. It shows how the $T_e$ distribution evolves with increasing $V_b$ from 2.0 to 8.0 V in a similar but slightly shorter device than the one used for Fig. 1. A comparison of Fig. 2c, d demonstrates that the peaks at the entrance and at the exit of the constriction, also identified by different sizes, swap places when the bias is reversed. Additional experiments on other devices with differing channel lengths (200 nm–1.0 μm) make it clear that (i) the $T_e$ profile generally exhibits a double-peaked structure, with the first hot spot within the channel close to the entrance and the second one outside the channel 100–250 nm away from the exit and that (ii) the first hot spot does not produce the corresponding signature in the $T_L$ profile (see Supplementary Note 3).

The one-dimensional profiles of $T_e$ and $T_L$ taken along the channel (in the $x$ direction at $y = 0$) in the device of Fig. 1 are plotted together for comparison in Fig. 3a. The upper inset replots the 2D image of $T_e$ in the middle panel in Fig. 1a on the same $x$-axis scale as that of Fig. 3a–d. The second $T_e$ peak outside the channel exit coincides with the single $T_L$ peak, but the first $T_e$ peak does not have a corresponding structure in the $T_L$ profile. Without theoretical interpretation, this feature implies that hot electrons heated in the vicinity of the channel entrance are thermally isolated from the lattice and that electron heating does not lead to discernible lattice heating. In addition, if the excess energy from hot electrons is dissipated during the travel through the channel, it would increase the left-hand side of the $T_L$-profile in Fig. 3a, making the curve asymmetric about the peak. The $T_L$ profile is, however, nearly symmetric around the peak outside the channel exit, indicating no discernible energy dissipation throughout the channel (see Supplementary Fig. 3 and Supplementary Note 3). Hence, the coincident peaks of $T_e$ and $T_L$ outside the channel exit suggest a surprising aspect that hot electrons are thermally isolated from the lattice throughout the

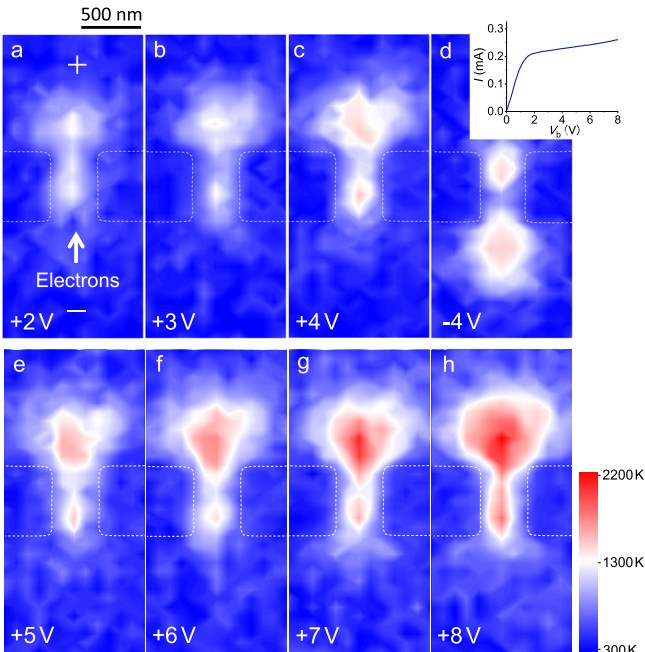

**Fig. 2 Evolution of the two-$T_e$-peak structure in the real-space hot-electron distribution with increasing bias voltage, $V_b$. a–h** 2D images of $T_e$ obtained with SNoiM, where **d** shows the image in the opposite bias polarity. The device studied is similar to that for Fig. 1, with a slightly different geometry of the constriction channel of 450 nm width and 640 nm length. White dotted lines are guides for the eyes to show the mesa edges that define the constriction. The current–voltage trend is shown in the inset.

channel and that the energy gained by the electrons passes through the channel without undergoing a significant loss and is eventually released to the lattice outside the channel exit. The length for which this quasiadiabatic transport takes place (roughly the channel length) largely exceeds the distance that hot electrons drift during the conventional energy relaxation time (see "Methods"). Hence, the phenomenon here requires explanation.

In the low-bias regime with $V_b = 3.0$ V, Fig. 3b shows that both $T_e$ and $T_L$ form roughly symmetric profiles centred at the middle of the channel ($x = 0$), reaching the peak values $T_e \approx 1200$ K and $\Delta T_L \approx 0.1$ K without exhibiting peculiar features.

**Hot LO-phonon bottleneck in the electron transport.** It is established that hot electrons in GaAs primarily dissipate energy by emitting longitudinal–optical (LO) phonons with energy $\hbar\omega_{LO} \approx 37$ meV at a typical rate of $1/\tau_{LO} = (0.12 \text{ ps})^{-1}$ (ref. [33]). The emitted LO phonons, in turn, are known to decay into two longitudinal acoustic (LA) phonons via the so-called Klemens channel after a decay lifetime of $\tau_{l,LO} \approx 5.0$ ps[34]. The two LA phonons eventually decay into thermalized longer-wavelength ($\lambda$) acoustic phonons, ending up with lattice heating. In the ordinary framework of hot-electron transport, possible rise in the effective LO-phonon temperature, $T_{LO}$, caused by the emitted LO phonons is supposed to be insubstantial.

In the present work, $T_{LO}$ can be significantly elevated because (i) the electron density is relatively high, (ii) the LO phonons do not spatially diffuse due to nearly vanishing group velocity, and (iii) the emitted LO phonons are of small wave numbers being confined within a narrow $q$-space sphere around the Brillouin zone centre, containing only a limited number of states (NOS = $4.00 \times 10^{18}/\text{cm}^3$)[35] (see Supplementary Fig. 5 and Supplementary

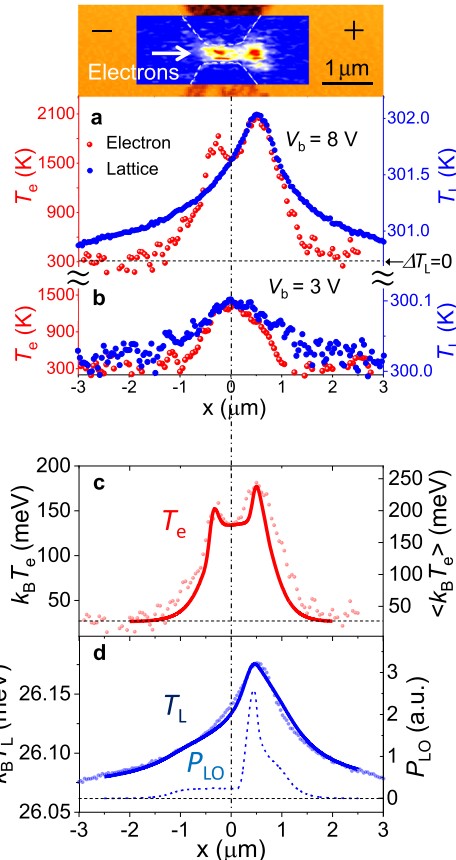

**Fig. 3 Comparison between electron heating ($T_e$) and lattice heating ($T_L$).**
**a** 1D profiles of $T_e$ (red dots, taken with SNoiM) and $T_L$ (blue dots, taken with SThM) taken along the device of Fig. 1 at $y = 0$ with a bias voltage $V_b$ = 8.0 V. The broken horizontal line shows the baseline for both $T_e$ and $T_L$ ($\Delta T_e = \Delta T_L = 0$ or $T_{base} = T_e = T_L = 300.8$ K), where the vertical scales are given so that the peak heights agree to one another. **b** 1D profiles of $T_e$ and $T_L$, similar to the data of **a** except that $V_b$ = 3.0 V. **c** Theoretically derived curve of $<T_e>$ (red line) for $V_b$ = 8.0 V reproduces the experimental $T_e$-profile with the two-peak structure (red dots). The broken horizontal line marks the position of $k_B T_{base}$ with $T_{base}$ = 300.8 K. **d** Theoretically derived single peak structure of $T_L$ (blue line) reproducing the experimental profile (blue dots). The blue dotted curve represents the theoretically derived rate of energy loss due to LO-phonon scattering, $P_{LO}$. The broken horizontal line marks the position of $k_B T_{base}$ with $T_{base}$ = 300.8 K for $k_B T_L$ and zero for $P_{LO}$.

Note 4). If the electrical input power, $p = Ej$, is dissipated utterly via the emission of LO phonons, the rate of LO-phonon emission is given by $\partial N_{LO}/\partial t = p/\hbar\omega_{LO}$. The emitted LO phonons, in turn, pile up to the nonequilibrium density higher than the thermal equilibrium value by $N_{LO} = (dN_{LO}/dt) \times \tau_{l,LO} = (p/\hbar\omega_{LO})\tau_{l,LO}$. One can hence estimate $T_{LO}$ through

$$(p/\hbar\omega_{LO})\tau_{l,LO} = \text{NOS}\{n_{LO}(T_{LO}) - n_{LO}(T_{Room})\},$$

where $n_{LO}(T) \equiv \{\exp(\hbar\omega_{LO}/k_B T) - 1\}^{-1}$ is the LO-phonon occupation number at temperature $T$ with the Boltzmann constant $k_B$. From this relation, $T_{LO}$ is explicitly derived to be

$$T_{LO} = (\hbar\omega_{LO}/k_B)/\ln[1 + \{Ap + n_{LO}(T_{Room})\}^{-1}] \quad (1)$$

with $A \equiv \tau_{l,LO}/(\text{NOS } \hbar\omega_{LO}) \approx 2.11 \times 10^{-10}$ cm$^3$/W. The blue line in Fig. 4 shows the values of $T_{LO}$ as a function of $p$ according to Eq. (1), which converge to the linear asymptotic form $T_{LO} = \{(\tau_{l,LO}/k_B)\text{NOS}\}p$ in the higher $p$ range.

Two black dots in Fig. 4 mark the experimental values of $T_e$, respectively, in the higher- and the lower-bias conditions

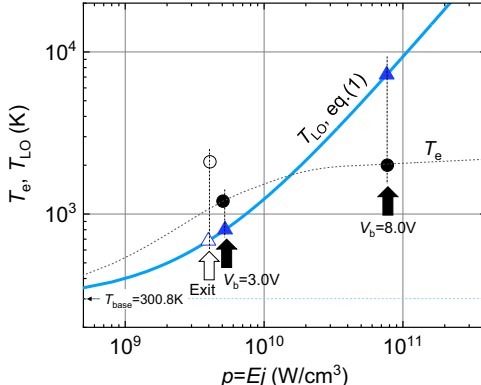

**Fig. 4 Criterion for determining the hot-phonon bottleneck effect.** The blue line shows $T_{LO}$ estimated from Eq. (1) against the electrical input power density $p$. The two black circles mark the experimentally determined effective electron temperatures in the conductor channel under two bias conditions (Fig. 3a); viz., $T_e = 2000$ K at $V_b = 8.0$ V and $T_e = 1200$ K at $V_b = 3.0$ V. The white circle marks the experimental value, $T_e = 2100$ K, outside the channel near the channel exit at $V_b = 8.0$ V. The criterion for the hot-phonon bottleneck regime is given by condition (2), $T_e < T_{LO}$. The thin black dotted line is a guide for eyes speculating possible variation of $T_e$ with $p$ in the channel.

($T_e = 2000$ K at $V_b = 8.0$ V and $T_e = 1200$ K at $V_b = 3.0$ V in Fig. 3a, b), where $V_b = 8.0$ V ($I = 0.25$ mA) and $V_b = 3.0$ V ($I = 0.145$ mA) correspond, respectively, to $p = Ej \approx 7.72 \times 10^{10}$ W/cm$^3$ ($E \approx 40$ kV/cm, $j \approx 1.93 \times 10^6$ A/cm$^2$) and $p \approx 5.06 \times 10^9$ W/cm$^3$ ($E \approx 4.5$ kV/cm, $j \approx 1.12 \times 10^6$ A/cm$^2$) in the channel (see Supplementary Note 5). It is noted that in the high-bias condition ($V_b = 8.0$ V) the LO-phonon temperature, $T_{LO} = 7,300$ K (blue triangle), estimated from Eq. (1) is distinctly higher than $T_e \approx 2000$ K. Differently, $T_{LO} = 780$ K (blue triangle) estimated from Eq. (1) in the low-bias condition ($V_b = 3.0$ V) is lower than $T_e \approx 1200$ K.

In the high-bias condition, where

$$T_e < T_{LO} \quad (2)$$

holds for $T_{LO}$ given by Eq. (1), the electron system would gain energy from the LO-phonon system through LO-phonon absorption. Since $T_{LO}$ is thereby reduced through the LO-phonon absorption, the value of $T_{LO}$ given by Eq. (1) is not physically realized. Self-consistent value of $T_{LO}$ has to be determined by considering both the emission and the absorption of LO phonons, and is shown to be nearly equal to $T_e$, in fact, slightly lower than $T_e$ (see Supplementary Fig. 7 and Supplementary Note 6). If inequality relation (2) holds, therefore, the net energy loss is significantly suppressed with $T_{LO} \approx T_e$, and the electron system is thereby adiabatically isolated from the lattice. This effect, which we call the hot-phonon bottleneck, accounts for the quasiadiabatic feature of the electron transport experimentally observed in the high-bias condition (Figs. 1a, b and 3a). In the lower-bias condition, where $T_e > T_{LO}$, the energy dissipation via LO-phonon emission is not significantly hindered, resulting in the ordinary hot-electron transport. This is consistent with the observed coincident broad symmetric peaks of $T_e$ and $T_L$ occurring in the middle of the channel (Fig. 3b).

Detailed profiles of $T_e$ and $T_L$ in the high-bias condition (Fig. 3a, $V_b = 8.0$ V) are interpreted below by considering spatial variation of $E$ (or $p$) around the channel (see Supplementary Fig. 6). The first peak of $T_e$ near the channel entrance occurs within the channel where condition (2) holds. Physically, the peak is ascribed to the well-known velocity overshoot (or overheating) of electrons caused by abruptly increasing electric fields near the

entrance[36,37]. Condition (2) is unaffected by the relatively small effect of electron overheating, which is roughly a 10% effect in amplitude. Hence the electron system remains adiabatically isolated from the lattice and the effect of electron overheating (or the first peak in $T_e$) does not cause any discernible signature in the profile of $T_L$. The other prominent peak of $T_e$ near the channel exit occurs outside the channel, where the electric field $E$ is distinctly lower such that $E \approx 4.0$ kV/cm, $j \approx 1.02 \times 10^6$ A/cm$^2$ and $p \approx 4.08 \times 10^9$ W/cm$^3$ (see Supplementary Fig. 6 and Supplementary Note 5). Equation (1) predicts $T_{LO} \approx 690$ K as marked by the white triangle in Fig. 4, while experimentally found $T_e \approx 2100$ K (white circle in Fig. 4, taken from Fig. 3a) is distinctly higher. Since condition (2) breaks down with $T_e \gg T_{LO}$, the energy dissipation via LO-phonon emission becomes possible. The excess electron energy stored via the adiabatic passage through the channel is released outside the channel, causing the single peak of $T_L$ that is coincident with the second $T_e$ peak (Fig. 3a). The physical mechanism of $T_e$ taking the largest peak outside the channel is that hot electrons released from the channel outlet drift over the energy relaxation length, reaching the outside region of low electrostatic potential. (This effect was discussed in our previous work as the non-local energy dissipation[23].) Unlike $T_e$, the value of $T_{LO}$ drops rapidly outside the channel exit causing $T_e \gg T_{LO}$.

The quasiadiabatic hot-electron transport discussed here can be expected to occur in a wide variety of materials because the linear increase in $T_{LO}$ with increasing $p$ like Eq. (1) and the sublinear dependence of $T_e$ on $p$ are supposed to be a general trend in high electric field transport in many conductors with appropriate interaction between charge carriers and LO phonons. There have been few reports, however, probably because of the lack of measurements so far. Aside from the electron transport, hot-phonon bottleneck effect has been extensively studied for the photoexcited transient state of III–V ionic crystals[24,25] and perovskite compounds[26–29], where the energy loss of photocarriers is found to significantly slow down at high excitation levels. In hot-electron transport phenomena, hot-phonon generation was reported experimentally in standard/exotic semiconductors[38,39,40–42], but its effect on the hot-electron kinetics has been left unclear. In the transport phenomena, theoretical discussion has been limited to the drift velocity of electrons with major concern about the possible degradation of device performance due to reduction in the electron mobility[1,2,43,44]. In contrast to the earlier efforts, direct visualization of $T_e$ and $T_L$ in real space at the nanoscale in this work has disclosed a quasiadiabatic electron transport by clarifying the phenomena from the viewpoint of energy transport.

In graphene nanoconstrictions, asymmetric $T_L$-profiles have been found in the measurements of SThM and interpreted in terms of Peltier effect[45,46]. In our experiments Peltier effect is ruled out because the local heating and the local cooling would take place, respectively, at the channel entrance and the channel exit, which is opposite to the observation in the present experiment (see Supplementary Fig. 3). It is, nevertheless, interesting to estimate the thermoelectric power due to Peltier effect in our experiment by assuming that the bulk Seebeck coefficient, $S_{bulk} \approx -100$ μV/K, in the wide lead region in our n-GaAs device[47] reduces to $S_{channel} = 0$ in the constriction channel. Since the heat flow $\dot{Q} = S_{bulk}TI = 7.5$ μW at $T = 300$ K at $I = 0.25$ mA ($V_b = 8.0$ V) is blocked at the constriction, the heat power $\dot{Q}_{Peltier} = 7.5$ μW is generated or annihilated at the entrance and the exit of the channel. This power is less than one per cent of the electrical input power $P_{Joule} = V_{channel}I = 2$ mW ($V_b = 8.0$ V) in our experiments. Hence the small amplitude of Peltier effect with respect to the $E$-induced Joule heat power is consistent with

the experimental observation. The ratio of the two powers is roughly given by $\dot{Q}_{Peltier}/P_{Joule} \approx S_{bulk}T/V_{channel} \propto S_{bulk}/\rho_{channel}$ with $\rho_{channel}$ the electrical resistivity of the constriction channel. While the amplitude of bulk Seebeck coefficient $S_{bulk}$ is similar between the two material systems, the distinct difference is the resistivity $\rho_{channel}$, which is by more than two orders of magnitude higher in the present GaAs constriction than in graphene constriction. In existing studies of nanoscale $T_L$-distribution[19,45,46], the $T_L$-profile is often divided into symmetric and antisymmetric parts with respect to the bias current polarity, and the former and the latter are interpreted, respectively, as due to Joule heating effect and Peltier effect. Unlike those existing studies, the present experiments disclosed, by simultaneously measuring $T_e$, that the antisymmetric part of the $T_L$-profile (see Supplementary Fig. 3) is entirely dominated by the $E$-induced Joule heat effect in the hot-electron condition.

**Hot-phonon bottleneck effect in the two-carrier transport.** In the high $E$ region exceeding $E_c \approx 10$ kV/cm, two-carrier transport is involved because hot electrons in GaAs transfer to upper X valleys, lying $\Delta\varepsilon_{\Gamma X} \approx 550$ meV above the $\Gamma$-valley (see Supplementary Note 7)[48,49]. Figure 5 illustrates schematically the kinetics of hot electrons interacting with hot phonons. While fundamental framework of the hot-phonon bottleneck effect is substantially unaffected, the electron–phonon kinetics are elaborated in more detail by explicitly considering the upper-valley transfer of electrons. The effective mass and the density of states of electrons in the X valleys are much larger than those in the $\Gamma$ valley, so that the $\Gamma \rightarrow X$ transfer significantly reduces the electron mobility, introducing sublinear dependence in the current vs. voltage characteristics, as seen for $V_b > 5.0$ V/cm in the bottom panel in Fig. 1c and for $V_b > 3.0$ V/cm in the upper right inset in Fig. 2.

The electron temperature probed with SNoiM in the two-carrier condition is assumed to be the mean electron temperature defined by

$$<T_e> = \frac{n_\Gamma}{n} \cdot T_\Gamma + \frac{n_X}{n} \cdot T_X, \qquad (3)$$

where $n_\Gamma(x)$, $n_X(x)$, $T_\Gamma(x)$ and $T_X(x)$ are the fractional densities and the effective temperatures of the electrons in $\Gamma$- and X valleys, respectively. Here, we assume $n = n_\Gamma(x) + n_X(x)$ to be a constant equal to the total electron density ignoring minor contribution from the L valleys[48]. The rate of net energy loss due to LO-phonon scattering is given by $P_{LO}(x) = P_\Gamma(x) + P_X(x)$, where

$$\begin{aligned} P_i = \frac{\Delta\varepsilon}{\tau_{LO}D_0} \int_0^\infty &\mathrm{d}\varepsilon D_i(\varepsilon)D_i(\varepsilon + \Delta\varepsilon) \\ \times [f_i(\varepsilon + \Delta\varepsilon) \cdot \{1 - f_i(\varepsilon)\} &\cdot (n_{LO} + 1) - f_i(\varepsilon) \\ \cdot \{1 - f_i(\varepsilon + \Delta\varepsilon)\} &\cdot n_{LO}] \end{aligned} \qquad (4)$$

for $i = \Gamma$ or X is a function of $x$ through $n_\Gamma(x)$ or $n_X(x)$, $T_\Gamma(x)$, $T_X(x)$ and $T_{LO}(x)$, and takes account of both the emission and the absorption of LO phonons ($\Delta\varepsilon = \hbar\omega_{LO}$) in each set of valleys. Here, $D_i(\varepsilon)$ is the density of states in each valley; viz., $D_\Gamma(\varepsilon) = (2^{1/2}/\pi^2\hbar^3)$ $(m^\Gamma_d)^{3/2}\varepsilon^{1/2}$, $D_X(\varepsilon) = (2^{1/2}/\pi^2\hbar^3)(m^X_d)^{3/2}(\varepsilon - \Delta\varepsilon_{\Gamma X})^{1/2}$ for $\varepsilon > \Delta\varepsilon_{\Gamma X}$ and $D_X(\varepsilon) = 0$ for $\varepsilon < \Delta\varepsilon_{\Gamma X}$ with the respective density-of-state effective masses $m^\Gamma_d = 0.067m_0$ and $m^X_d = 1.09m_0$. $D_0 = 3.24 \times 10^{25}$(m$^3$eV)$^{-1}$ is a constant describing the average density of states of hot electrons. The distribution function of electrons in each valley, $f_i(\varepsilon) = 1/[\exp\{(\varepsilon - \mu_i)/k_BT_i\} + 1]$, is characterized by $T_i(x)$, where the local electrochemical potential $\mu_i(x)$ is determined by $n_i(x) = \int_0^\infty f_i(\varepsilon)D_i(\varepsilon)\mathrm{d}\varepsilon$. The LO-phonon occupation number, $n_{LO} \equiv \{\exp(\hbar\omega_{LO}/k_BT_{LO}) - 1\}^{-1}$, is determined by $T_{LO}(x)$.

Values of $<T_e(x)>$ and $P_{LO}(x)$ in the device shown in Figs. 1 and 3 at $V_b = 8.0$ V can be estimated by speculating values of

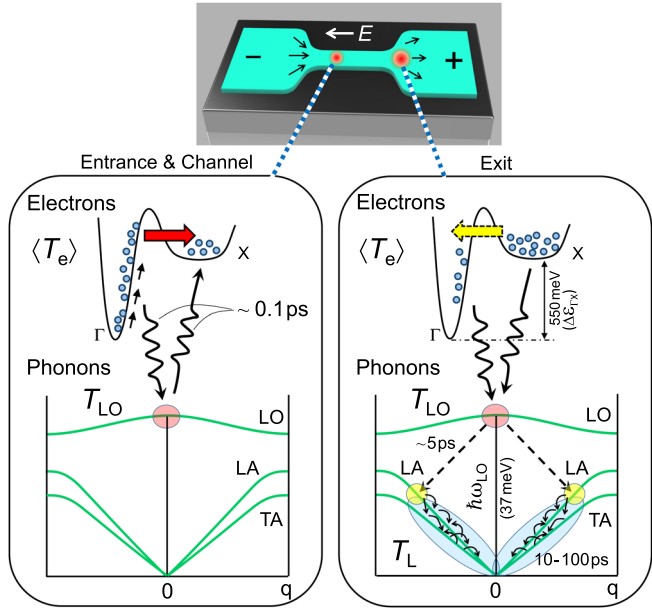

**Fig. 5 Schematic representation of the kinetics of hot electrons and hot phonons.** As electrons enter a narrow constriction channel, they are rapidly accelerated by intense electric field $E$ and the effective electron temperature $<T_e>$ is elevated, as studied with SNoiM. It causes frequent LO-phonon emission in the Γ-valley (down-pointing wavy arrow in the left column) as well as Γ → X intervalley electron transfer (fat red arrow in the left column). The effective LO-phonon temperature, $T_{LO}$, is significantly elevated, but X-valley electrons are not efficiently heated by $E$, so that the LO phonons emitted by Γ-valley electrons are strongly absorbed by X-valley electrons (wavy arrow pointing upward in the left column), suppressing the net energy loss in the channel and causing the "hot-phonon bottleneck effect". As the electrons exit the channel, Γ-valley electrons are no longer rapidly accelerated by $E$, but energy is still fed via intervalley back-transfer from X valleys (fat yellow arrow in the right column). $T_{LO}$ drops faster than $T_X$ so that X-valley electrons change to emit LO phonons (wavy arrow pointing downward in the right column). The electrons dissipate net energy, causing prominent non-local energy dissipation near the exit. Emitted LO phonons quickly decay into two LA phonons (via Klemens channel), eventually thermalize slowly into heat, and are primarily composed of long wavelength acoustic phonons that are sensed with SThM as the lattice temperature $T_L$.

$T_Γ(x)$, $T_X(x)$, $n_Γ(x)$, $n_X(x)$ and $T_{LO}(x)$ based on the Monte Caro simulation, where relevant quantities are derived against $E$[48] (see Supplementary Fig. 8 and Supplementary Note 3). For the estimation, we take into account the effect of finite channel length[36,37] considering the spatial distribution of $E(x)$ in the device (see Supplementary Fig. 6). We also note that $n_X$, $T_Γ$ and $T_X$ in the channel take larger values due to hot-phonon bottleneck effect.

In the channel ($E ≈ 40$ V/cm), the majority of electrons are expected to transfer to X valleys with elevated electron temperatures in respective valleys; viz., $n_X/n ≈ 0.83$ ($n_Γ/n ≈ 0.17$), $T_Γ ∼ 3250$ K ($k_B T_Γ ∼ 280$ meV) and $T_X ∼ 1740$ K ($k_B T_X ∼ 150$ meV) (see Supplementary Figs. 8a, b). Here, $T_X$ is substantially lower than $T_Γ$ because the electron mobility in X valleys is much lower. As discussed in the last section, the hot-phonon bottleneck effect makes $T_{LO}$ in the channel close to but slightly lower than $T_e$, which implies, in the two-carrier condition, that $T_{LO}$ is lower than $T_Γ$ but slightly higher than $T_X$; viz., $T_{LO} ∼ 1750$ K (see Supplementary Fig. 8c). The left panel of Fig. 5 schematically depicts the hot-phonon bottleneck effect in the two-carrier condition, where rapidly accelerated Γ-valley

electrons frequently emit LO phonons ($T_Γ > T_{LO} ⇒ P_Γ > 0$) elevating $T_{LO}$, while less hot X-valley electrons absorb the emitted LO phonons ($T_X < T_{LO} ⇒ P_x < 0$), nearly cancelling the loss of energy ($P_{LO} = P_Γ + P_X ∼ 0$). Hence, quasiadiabatic electron transport through the channel is realized by storing the kinetic energy acquired by Γ-valley electrons in the upper X valleys ($Δε_{ΓX} ≈ 550$ meV).

Detailed structures in the profile of $<T_e(x)>$ and $P_{LO}(x)$ arise in connection with nonstationary conditions of the transport caused by the rapidly varying $E(x)$, as discussed in the next paragraph. In Fig. 3c, a solid red line shows that theoretical values of $<T_e(x)>$ reproduce well the experimentally observed profile of $T_e$, including the double-peak structure. A dotted blue line in Fig. 3d shows that theoretical values of $P_{LO}(x)$ are suppressed in the channel but take a prominent peak outside the channel close to the exit (see Supplementary Fig. 8d), demonstrating the hot-phonon bottleneck effect. The profile of the lattice temperature $T_L(x)$ is broadened due to the lattice thermal conduction[32], and is theoretically derived from $P_{LO}(x)$ by assuming a symmetric broadening parameter (see Supplementary Note 9). A solid blue line in Fig. 3d shows that the theoretically derived profile of $T_L(x)$ well reproduces the experimentally found single broad peak of $T_L$ outside the channel exit.

Detailed electron kinetics causing the profile of $<T_e(x)>$ and $P_{LO}(x)$ described in the last paragraph are discussed below (see Supplementary Figs. 8a–d). When Γ-valley electrons approach and enter the channel (from the left-hand side of the device depicted in the upper column of Fig. 5), the increase of the average kinetic energy of electrons is suppressed by the Γ → X transfer. Since the intervalley transfer slightly delays by the intervalley scattering time (roughly 40 fs) compared to the acceleration by $E$, the suppression of energy delays near the channel entrance where $E$ rapidly increases (see Supplementary Fig. 5), resulting in an overheating/over-population of Γ-valley electrons; that is, values of $T_Γ$ and $n_Γ$ are slightly larger than the steady-state values expected from the local electric field $E(x)$ near the entrance. This causes a peak of $<T_e(x)>$ near the entrance. After adiabatically transmitted through the channel, Γ-valley electrons ($n_Γ/n ≈ 0.17$, $T_Γ ∼ 3250$ K) and X-valley electrons ($n_X/n ≈ 0.83$, $T_X ∼ 1740$ K) are released from the channel exit to the wider lead region (Fig. 5), where hot-phonon bottleneck effect is extinguished with distinctly lower $E$ and $j$ (white arrow in Fig. 4). Hot Γ-valley electrons readily spread to the outside region close to the exit ($x ≈ 600$ nm, $E ≈ 4$ kV/cm) within the energy relaxation time ($∼1$ ps)[23]. Meantime, X-valley electrons rapidly back-transfer to the Γ-valley ($∼40$ fs) as schematically illustrated in the right panel of Fig. 5. It follows that hot electrons are efficiently supplied to the Γ-valley from the X valleys (right panel of Fig. 5), whereas Γ-valley electrons no longer rapidly gain energy from $E$ so that $T_{LO}$ falls lower than $T_X$. The rate of LO-phonon emission by Γ-valley electrons is thereby maintained to be high. Due to the back-transfer, X-valley electrons quickly disappear near the channel exit ($n_{X/n} → 0$ and $n_Γ/n → 1$), making the LO-phonon reabsorption insubstantial and thereby promoting the onset of net LO-phonon emission. The coincident peaks of $T_e$ and $T_L$ accordingly occur immediately outside the channel exit. Briefly, the hot-phonon bottleneck effect is lifted when the electrons leave the channel, the energy stored in the X valleys for adiabatic transmission is returned to the Γ-valley and dissipated to the lattice.

The suppression ratio of the energy loss rate, $γ_{supp} = P_{LO}/P_0$, defined by the ratio of $P_{LO}$ to the fictitious loss rate $P_0 ≡ P_{LO}(T_{base})$ expected in the absence of hot-phonon effect ($T_{LO} = T_{base} = 300.8$ K), is about 1% in the channel: Similarly, the profile of $P_{LO}(x)$ in Fig. 3d suggests that approximately 93% of the energy gained from $E$ is transmitted through the channel without dissipation (see Supplementary Fig. 9 and the discussion in Supplementary Note 8).

This work has experimentally demonstrated a unique approach to access energy transport by probing different effective temperatures of nonequilibrium subsystems, which proved to be powerful for understanding the physics of current-carrying narrow conduction channels. In narrow GaAs constriction channels at high electric fields, conduction electrons generate LO phonons with a high density, while the emitted dense LO phonons prevent efficient cooling of hot electrons, giving rise to quasiadiabatic electron transport over a long distance around 1 μm at room temperature. The knowledge obtained here can serve as a building block for innovative on-chip energy management and energy-harvesting technologies.

## Methods

**SNoiM and estimation of $T_e$.** The instrument is a home-built microscope. The spatial resolution is ~50 nm, which is primarily determined by the probe tip. The principle and the construction of SNoiM are described in refs. [23,30] (see also Supplementary Note 1). SNoiM exclusively senses evanescent radiation localized on the material surface, but does not sense the familiar THz photon emissions such as those due to the blackbody radiation[50], externally induced coherent electron motion[51], and the one-particle radiative transition between the initial and the final states[52]. This is because all those photon emissions do not yield evanescent field on the material surface. Detected with SNoiM is the charge/current fluctuation that generates intense evanescent waves but cancels out in the region away from the surface. In this work it is the hot-electron shot noise, the intensity of which is most simply characterized by the effective electron temperature $T_e$. Absolute values of $T_e$ are derived from the signal intensity without using any adjustable parameter (see Supplementary Note 3).

SNoiM is thus far the only instrument that visualizes hot electrons in the steady-state transport condition, whereas in the photoexcited transient condition, hot electrons have been imaged by utilizing plasmonic techniques[53].

**SThM and estimation of $T_L$.** A commercial SThM (ANASYS INSTRUMENTS, NanoTA) is used to map the local lattice temperature distribution. A nanoscale temperature-sensitive resistive element is attached to the apex of an AFM tip, which is scanned across the sample surface in contact mode. The resistance change is measured with a Wheatstone bridge circuit, and the output voltage is referred to as the SThM signal. By scanning the surface of a well-calibrated pt100 planar resistive thermometer self-heated to a known temperature, we establish the SThM transfer characteristic linking its signal to the $T_L$ of the sample under study. We note that despite this SThM calibration procedure, the real local lattice temperature of a particular sample may differ from the readings due to a number of mechanisms[19], particularly when operated in air, so the absolute values have a significant uncertainty, but the spatial distribution of the temperature is unaffected. The spatial resolution of the equipment is nominally 20 nm, but the realistic resolution is supposed to be ~50 nm in the present experiment made in the ambient condition (see Supplementary Note 2).

The temperature measured with SThM is the lattice temperature $T_L$ because heat flow is dominated by the lattice that has a heat capacity several orders of magnitude larger than that of conduction electrons.

**GaAs/AlGaAs QW structure, devices and transport coefficients.** The GaAs/AlGaAs heterostructure used in this work is similar to the one described in ref. [23], which was grown with molecular beam epitaxy on the (100) plane. A quasi 2D electron gas (2DEG) layer with a density $n_{2D} = 1.16 \times 10^{13}$ cm$^{-2}$ or $n = 3.3 \times 10^{18}$ cm$^{-3}$ (corresponding to the Fermi energy $E_F = 119$ meV at absolute zero temperature $T = 0$ K) and Hall mobility $\mu = 0.167$ m$^2$/Vs is provided in a $W = 35$-nm-thick GaAs QW located 13 nm below the surface. The devices studied are fabricated with standard electron beam lithography and wet mesa etching with a depth ~100 nm. The constriction channel is connected to the source and drain contacts through 2DEG leads with a typical width of ~20 μm and a total length of ~190 μm. The effective voltage applied to the short constriction channel, $V_{channel} = V_b - V_{leads}$, is less than the bias voltage $V_b$ by the voltage drop along the leads $V_{leads}$, which depends on the device-specific accurate dimensions of the leads. The electric field $E$ in each device is evaluated by considering the known device-specific lead geometry (see Supplementary Note 5). Ohmic contacts of the source and the drain are prepared by alloying with AuGeNi. The drift velocity of electrons is experimentally estimated to be $v_d = I/(Wn_{2D}e) \approx 3.5 \times 10^4$ ms$^{-1}$ from $I = 0.25$ mA in the device with $V_b = 8.0$ V shown in Figs. 1 and 3. Hence, the distance the hot electrons drift in the high electric field during an event of LO-phonon scattering, $L_{d,LO} = v_d\tau_{LO} \approx 4.2$ nm with $\tau_{LO} = 0.12$ ps[33], is far smaller than the length scales of the channel.

## Data availability

The data that support the findings of this study are available from the corresponding authors upon reasonable request.

## Code availability

The source code to run all analyses in this paper can be obtained from the corresponding authors upon reasonable request.

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

## Acknowledgements
Z.A. and W.L. acknowledge financial support from the Shanghai Science and Technology Committee under grant nos. 18JC1420400, 18JC1410300, 20JC1414700 and 20DZ1100604; the National Natural Science Foundation of China (NSFC) under grant nos. 12027805, 61521005, 11991060, 11634012 and 11674070; and the National Key Research Program of China under grant no. 2016YFA0302000. Q.W. acknowledges the support in part by Grants-in-Aid for JSPS Fellows, International Secondment Scheme of the UK National Measurement System, and Grants-in-Aid for Scientific Research (B) No. 20H01846. S.K. acknowledges the support by the Chinese Academy of Sciences Visiting Professorships for Senior International Scientists. A.T. acknowledges the support of the UK government Department for Business, Energy and Industrial Strategy. The authors thank C. Barton and R. Puttock for assistance with the SThM measurements, and Weikang Lu and Wei Li for band structure calculation. Part of the experimental work was carried out in the Fudan Nanofabrication Laboratory.

## Author contributions
Q.W., S.K., Z.A. and A.T. conceived the idea and designed the experiments. Z.A. and L.Y. first observed the two-peak structures for $T_e$. Q.W. and L.Y. collected the SNoiM data presented in this work. Q.W. performed the SThM measurements. Q.W., L.Y., Z.A. and S.K. performed the data analysis. P.C. grew the crystal wafers for samples. Q.W., Z.A., A.T. and S.K. co-wrote the manuscript with comments from all authors. Z.A. and W.L. co-supervised the project in China.

## Competing interests
The authors declare no competing interests.
