## [Peer Review File · Nature Communications]

REVIEWER COMMENTS

Reviewer #1 (Remarks to the Author):

In this work the authors report a low loss transmission of energy carried by hot electrons in narrow conducting channels. They highlight a quasiadiabatic transport over distances hundred folds larger than the mean-free path of electrons by making parallel measurement of electronic and lattice temperatures inside these channels. They relate this quasi-adiabatic transport to the so called hot-phonon bottleneck effect which has been largely described in the literature. However, here they describe how this effect which is well known to impact the relaxation dynamic of hot carriers play a role on their transport and on the dissipation mechanisms when they propagate inside narrow conducting channel. Beside their fundamental interest for a better understanding of relaxation mechanisms in solid state physics, the results reported in this work are very promising in the point of view of applications. Hence, they could contribute for developing new and more efficient (weakly dissipating) nanoelectric devices. They could also be useful in the development of a new generation of energy conversion devices working in non-equilibrium regime. For these reasons the present work could deserve publication in Nature Communication. However, several important questions must be addressed before considering it for publication.

-Explain why in specific regions some electrons seem to be much colder than the room temperature?

In particular on Fig. 1-a (bottom) we see that around $y = -0.5$ micron $T_e \sim 200K$!

-What are the measurement precision for the electrons and lattice temperatures? Show the error bars on the temperature profiles.

-What is the timescale to which the temperatures are recorded?

-At low bias voltage ($V_b = 3V$) the plots in Fig3-b show that the maxima of lattice temperature profiles coincides with the maxima of electrons temperature profiles. On the other hand this is not the case anymore when $V_b = 8V$. In this case only the peak of electronic temperature in the exit area coincide with the peak of lattice temperature meaning that the lattice is not heated by the electrons in the entrance. It is assumed that for large bias voltages the electrons in the entrance zone are insulated from the lattice. But the only difference between the small bias voltage and the larger ones is the value of the electron temperature. The authors should explain/comment why when the electrons are hot they are isolated from the lattice and why this occurs only in the entrance of the channel? A detailed modeling of heat transport through the channel could definitely help in clarifying and confirming the different mechanisms and assumptions by reproducing the experimental data.

-In the section "Hot LO-phonon bottleneck in the steady state electron transport" the authors describe the so called hot-phonon bottleneck effect without transport. Only the LO-phonon emission rate is estimated. But inside the channel density N_{LO} a priori depends on the position. To describe its evolution and highlight the role play by the transport on the bottleneck effect the Boltzmann's equation for N_{LO} should be solved inside the channel by taking into account the coupling mechanisms (i.e. the colliding terms) and a mapping of electric field should be added.

On its present form the modeling is too empirical to make the conclusions clear enough and unquestionable.

Reviewer #2 (Remarks to the Author):

The manuscript employs two complementary scanning probe techniques to probe energy dissipation in a III-V semiconductor device. In their first approach the authors probe evanescent fields by scattering radiation from a tip into a sensitive detector at a well chosen frequency and relate the intensity and

spatial variation of the scattered radiation to the temperature of hot carriers. In a second experiment performed using a commercial scanning thermal probe in air, the authors attempt to measure the lattice (i.e. phonon) temperature. Finally from the measured spatial maps the authors argue that they have identified a novel mechanism that reduces energy dissipation from electrons. Overall, while the work is interesting, I find that there are several assumptions in the work that make the arguments somewhat speculative and the final conclusions somewhat weak. Please find below my comments:

1) Key to the interpretation of the results is the assumption that equations S1 and S2 are applicable even when the electrons are under highly non-equilibrium conditions. This is hard to justify.

2) The SThM imaging is done in ambient conditions. Under such conditions conduction through air is expected to dramatically impact both the ability to do quantitative thermal imaging and the resolution of the measurements. Therefore, the SThM data is not very robust. In fact, as mentioned in the methods section, the height of the mesa structure on these devices is 100 nm. One would therefore expect significant temperature artifacts at the edges of the channel in SThM images, if the heat transfer is dominated by tip sample contact (see F. Menges, Nature Comm. (2016)). However, T_L does not show any artifacts and looks very continuous suggesting that there is significant averaging of temperature fields, possibly due to conduction through air.

3) For the data shown on Fig. 1 the images of T_e and T_L need to have a larger field of view. An increased field of view for T_L is crucial, since T_L could be just showing local cooling at the inlet of the channel due to Peltier or Thomson effect .

4) The estimated $T_L=300.8$ K does not seem to be consistent with a simple joule heating and thermal conductance calculation. Considering a substrate of GaAs with thickness of 250 μm and the shown geometry, and neglecting all the thermal contact resistances, a thermal resistance of 2.2×10^4 K/W can be obtained. This leads to a temperature rise of 44 K due to joule heating of 2 mW, which is much larger than that reported by the authors.

5) Finally, the mechanism proposed by the authors does not make much sense to me. The authors state that heat dissipation occurs at the channel exit as indicated by the SThM signal. This means that the electrons collectively lose energy at the channel exit. Yet the prominent peak of the T_e occurs at this location, why is this? Further, when optical phonons are emitted and reabsorbed one would expect that the momentum of the electrons will be randomized and lead to a substantial electrical resistance, which would imply a local drop in voltage, which is undesirable and not an effect that is technologically useful as suggested by the authors..

For all the reasons described above, I think the results described in the paper are not robust.

Reviewer #3 (Remarks to the Author):

Manuscript presents simultaneous experimental measurements of electronic and lattice temperatures in current-carrying GaAs/AlGaAs nanoelectronic devices. Authors argue that due to hot "LO phonon bottleneck" electrons pass through in quasiadiabatic manner (i.e. without net energy loss). Experimental results are compared with theoretical modeling of energy transfer between electronic and phonon subsystems. The analysis is based on thermal distribution and golden rule transfer rates assumptions. Ability to measure both temperatures simultaneously is really exciting. Thus, manuscript deserves to be published.

There are several points authors may want to address before the publication:

1. On p.4 of the manuscript authors discuss "hot-phonon bottleneck effect", where emission of energy from electrons is compensated by absorption of energy from hot LO phonons. It seems, that the effect is common for any steady-state. Indeed in steady-state phonons are already hot enough so that net energy exchange with electron subsystem is zero. A short explanation would be helpful here.

2. Related to the point above: discussion of improbability of absorption of the LO phonons at low bias seems to be equivalent to saying that steady-state between electrons and LO phonons cannot be achieved at low bias. Only high bias leads to not energy exchange zero. It is not clear why this is the case: steady state seem to be achievable at any bias.

3. On p.11: "SThM does not measure $T_{\{LO\}}$ nor $P_{\{LO\}}$..." Does it mean that if measurement of $T_{\{LO\}}$ was possible, one would get $T_{\{LO\}}=T_e$?

4. On p.5 of Supplementary Information authors state: "the electron energy distribution, in either valley, is expected to deviate substantially from the Fermi function... Due to these approximations, the discussion in this work may be subject to slight quantitative modification." This statement sounds strange: all the analysis (experimental and theoretical) in the work is based on assumption thermal distributions for electron and phonon subsystems. If these assumptions do not hold, one in principle cannot introduce concept of temperature and then all the analysis fails.

5. p.14 of Supplementary Information: "In these calculations of Ref.43... hot phonon distribution is not considered." Does it mean that possibility to reach steady-state between electron and phonon subsystems is removed from theoretical modeling? With mechanism of energy exchange between the two subsystems being the main point of the manuscript, removing it from theoretical simulations seems to be too drastic an approximation. A discussion would be very helpful here.

Authors' response to Reviewer #1

In this work the authors report a low loss transmission of energy carried by hot electrons in narrow conducting channels. They highlight a quasiadiabatic transport over distances hundred folds larger than the mean-free path of electrons by making parallel measurement of electronic and lattice temperatures inside these channels. They relate this quasi-adiabatic transport to the so called hot-phonon bottleneck effect which has been largely described in the literature. However, here they describe how this effect which is well known to impact the relaxation dynamic of hot carriers play a role on their transport and on the dissipation mechanisms when they propagate inside narrow conducting channel. Beside their fundamental interest for a better understanding of relaxation mechanisms in solid state physics, the results reported in this work are very promising in the point of view of applications. Hence, they could contribute for developing new and more efficient (weakly dissipating) nanoelectric devices. They could also be useful in the development of a new generation of energy conversion devices working in non-equilibrium regime. For these reasons the present work could deserve publication in Nature Communication. However, several important questions must be addressed before considering it for publication.

Reply: We thank Reviewer for his/her careful reading of our manuscripts and for affirmative comments to the significance of our work. As to the questions of Reviewer, we think to have thoroughly answered below, and believe to have accordingly improved the manuscripts in the revised version. We hope that our answers and the revision of our manuscript will satisfy Reviewer.

-Explain why in specific regions some electrons seem to be much colder than the room temperature? In particular on Fig. 1-a (bottom) we see that around $y=-0.5$ micron $T_e \sim 200K$!

Reply: In the region outside mesa (e.g., $y = -0.5 \mu\text{m}$, more precisely $0.2 \mu\text{m} < |y|$ in Fig. 1a bottom), there are no conduction electrons so that SNoiM signal vanishes (implying $T_{\text{Room}} \approx 300 \text{ K}$). Hence the apparent signal corresponding to $T_e \sim 200 \text{ K}$ at $y = -0.5 \mu\text{m}$ in Fig. 1a arises from low-frequency noise yielding the experimental uncertainty roughly $\Delta T_e \sim \pm 100 \text{ K}$ in the measurements of SNoiM here. While the noise can be reduced by taking longer measurement time for signal averaging, we choose 300 ms/pixel with one pixel = $50 \text{ nm} \times 50 \text{ nm}$ for the measurements, which makes reasonable compromise between the experimental accuracy and the measurement time. The signal fluctuation is significantly reduced in non-etched regions where true signals due to conduction electrons are present. Therefore, the experimental uncertainty, $\Delta T_e \sim 100 \text{ K}$, in the etched region does not substantially affect the quantitative discussion in this study. To avoid misunderstanding, we add brief description of the experimental uncertainty and the measurement time to the caption of Fig. 1a & b in the revised manuscript.

-What are the measurement precision for the electrons and lattice temperatures? Show the error bars on the temperature profiles.

Reply: The answer partly overlaps our response to the last question. The precision of T_e in the measurement of SNoiM is within $\Delta T_e = \pm 100$ K in the region outside mesa (without conduction electrons) and $\Delta T_e/T_e = \pm 7\%$ in the region of mesa (with conduction electrons). The relative precision of T_L in the measurement of SThM in given images is roughly $\Delta T_L \sim \pm 100$ mK. (As in all the existing SThM measurements made in the ambient condition, the absolute accuracy is difficult to quantify because SThM signals are affected by detailed unidentified conditions including air-mediated thermal conduction etc.: This is briefly noted at the end of Methods, “SThM and estimation of T_L ” in the original manuscript.) We wish to mention that the discussion in the present work is unaffected by these experimental uncertainties.

In response to the Reviewer’s suggestion to show error bars, we give additional description of the experimental accuracy of T_e and T_L in the revised captions of Fig. 1 a & b. We think this is more reasonable than error bars because the experimental uncertainty ΔT_e is dependent on the location and ΔT_L is different depending on whether it refers to the relative or the absolute accuracy.

-What is the timescale to which the temperatures are recorded?

Reply: The timescale is 300 ms/pixel (one pixel: 50×50 nm²) for the measurement of SNoiM (T_e) and ~ 8 ms/pixel (one pixel: 40×40 nm²) for the measurement of SThM (T_L). Accordingly, it takes ~ 9 minutes and ~ 0.4 minutes, respectively, for the 2D-images of T_e (SNoiM, Fig. 1a) and T_L (SThM, Fig. 1b). In the revised manuscript, we add this information to the figure captions.

-At low bias voltage ($V_b=3V$) the plots in Fig3-b show that the maxima of lattice temperature profiles coincides with the maxima of electrons temperature profiles. On the other hand this is not the case anymore when $V_b=8V$. In this case only the peak of electronic temperature in the exit area coincide with the peak of lattice temperature meaning that the lattice is not heated by the electrons in the entrance. It is assumed that for large bias voltages the electrons in the entrance zone are insulated from the lattice. But the only difference between the small bias voltage and the larger ones is the value of the electron temperature. The authors should explain/comment why when the electrons are hot they are isolated from the lattice and why this occurs only in the entrance of the channel? A detailed modeling of heat transport through the channel could definitely help in clarifying and confirming the different mechanisms and assumptions by reproducing the experimental data.

Reply: We greatly appreciate the questions of Reviewer, which are tied to the crucially important points of our work. Whereas those points were explained in the original version of the manuscript, the explanation may have been too compact for thorough understanding.

There are three points to be made. First, the difference between the high and the low bias conditions ($V_b = 8$ V and $V_b = 3$ V) is not only in the electron temperature ($T_e \approx 2,000$ K and $T_e \approx 1,200$ K), but also in the estimated effective LO-phonon temperature ($T_{LO} \approx 7,300$ K and $T_{LO} \approx 780$ K). Whether or not the electrons are adiabatically isolated from the lattice (due to the hot-phonon bottleneck effect) is determined not by T_e alone, but by the relative amplitude of T_e with respect to T_{LO} ; viz., the electrons are adiabatically isolated if $T_e < T_{LO}$, but not the case if $T_e > T_{LO}$. In the high bias condition ($V_b = 8$ V) adiabatic isolation occurs because $T_e < T_{LO}$, while in the low bias condition ($V_b = 3$ V) this is not the case because $T_e > T_{LO}$. This was explained in the section “*Hot LO-phonon bottleneck in the steady-state electron transport*” (on pages 7 & 8) in the original version of the manuscript.

Secondly, at $V_b = 8$ V the adiabatic isolation occurs not only at the entrance but throughout the whole conduction channel, because $T_e < T_{LO}$ is satisfied throughout the channel. Hence no structures of the T_e profile in the whole channel (including the entrance) give rise to discernible signatures in the T_L profile. This was stated by the sentence from the 5th line from the bottom through the bottom on page 6 in the original text.

Thirdly, the adiabatic isolation at $V_b = 8$ V breaks down when the electrons leave the constriction channel and the ordinary condition of hot electrons recovers ($T_e > T_{LO}$) outside the channel close to the channel exit: This gives rise to the coincident peaks of T_e and T_L outside the channel at $V_b = 8$ V. This was described in the section “*Semiquantitative analysis of electron transport with hot-phonon bottleneck*” (from page 8 through page 11) in the original text.

All the points in the above have been made clearer in the revised version. The section “*Hot LO-phonon bottleneck in the electron transport*” has been expanded to pages 7 -11 for explicit discussion of all the points. Particularly, T_{LO} is given with new Eq. (1) and the judgement condition of the hot-phonon bottleneck effect is explicitly given with new Relation (2). Figure 4 is added for better understandability. (To support the discussion, we also add, in Supplementary Information, “V. Estimation of E in the channel” and “VI. Self-consistent determination of T_{LO} in a simplified model”.) As to point 2, experimental aspect is described in detail from the 5th line from the bottom on page 6 through the 6th line from the top on page 7 in the revised main text. In the subsequent section of the main text, “*Hot-phonon bottleneck effect in the two-carrier transport*” from page 11 through page 14, we highlighted the consequence of electron transfer to upper X valleys, and elaborate the interpretation of the detailed fine structures in the profile of T_e and T_L . Particularly, point 3 is discussed in detail.

We believe that the comprehensive revision has made the manuscript distinctly clearer and transparent. We hence hope that Reviewer will be satisfied with the new version of the manuscript.

-In the section “Hot LO-phonon bottleneck in the steady state electron transport” the authors describe the so called hot-phonon bottleneck effect without transport. Only the LO-phonon emission rate is estimated. But inside the channel density N_{LO} a priori depends on the position. To describe its evolution and highlight the role play by the

transport on the bottleneck effect the Boltzmann's equation for N_{LO} should be solved inside the channel by taking into account the coupling mechanisms (i.e. the colliding terms) and a mapping of electric field should be added.

Reply: We thank Reviewer for asking about the general validity of our theoretical arguments. We feel that our compact discussion in the original version has led to some misunderstanding. We believe all the points have been made clear enough in the newly added descriptions of the revised version of the manuscripts, especially in the new sections “Hot LO-phonon bottleneck in the electron transport” and “Hot-phonon bottleneck effect in the two-carrier transport”. Below we address each point of Reviewer comments to explain that our theoretical analysis is practically the only viable and reliable approach for the present study.

Before proceeding to detailed description, we particularly appreciate his/her comment that a mapping of electric field E should be added. Following the suggestion, we have derived the profile of $E(x)$ in our device of Figs. 1 and 3 at $V_b = 8$ V via simulation calculation based on the finite-element method, and show it in Fig. S6 in the revised version. (Experimental mapping would be nicer but is technically a heavy challenge.) We also add a new section “*V. Estimation of E in the channel*” in Supplementary Information in the revised version to describe how to estimate E . The figure below is the replot of Fig. S6, in which the red broken line shows $E(x)$ strongly concentrating in the channel reaching a maximum value ~ 40 kV/cm: The value is consistent with the experimental estimation from the current voltage characteristics.

$E(x)$ in the device of Figs. 1 & 3 at $V_b=8$ V.

Replot of Fig. S6 in the revised Supplementary Information

As Reviewer suggests, the LO-phonon density N_{LO} (or T_{LO}), as well as other important physical quantities, i.e., n_{Γ} , n_X , T_{Γ} and T_X , certainly evolves as the electrons move in the channel. However, valid discussion can be made without detailed theoretical analysis of those position-dependent evolving quantities. This is because only the electric field intensity and the current density in the channel (in addition to the well-known kinetics of LO phonons) suffice for the discussion and these two quantities

can be reliably determined in the experiments. Hence the two key findings of this work, that (a) *hot electrons in the channel under high bias conditions are adiabatically isolated from the lattice* and that (b) *this is caused by the hot-phonon bottleneck effect*, are robust messages independent of detailed theoretical analysis. In more specific terms, (a) is a conclusion directly derived from the experimental images of T_e and T_L (as described in the revised main text from the 2nd line from the bottom of page 6 through the 6th line from the top of page 7), and the argument of (b) is based on the established knowledge of the LO phonon kinetics along with the electric field strength and the current density in the channel derived from the experimentally determined current voltage characteristics. Detailed analysis of the evolution of position-dependent quantities is thus not of the highest priority: It serves only to provide elaborated interpretation of the effect, including the detailed fine profiles of T_e and T_L in regions near the channel entrance and the channel exit, where E rapidly varies. For this detailed analysis, inclusion of two-types of electrons is indispensable (see section “*Hot-phonon bottleneck effect in the two-carrier transport*”, pages 11-15 in the revised manuscript).

We understand the Reviewer’s suggestion that the analysis should better be made in a rigorous theoretical framework, say, by solving Boltzmann transport equations. In practice, however, it is difficult to carry out. To obtain reliable results, *position-dependent* energy distribution functions of a number of physical quantities have to be self-consistently determined together with the *position -dependent* electric field $E(r)$. For this sake, a number of Boltzmann transport equations have to be solved simultaneously, which costs a huge computational domain with multiple spatial dimensions. Given available computational capabilities of practical computers, it is unrealistic to carry out such numerical calculation in a reliable manner. In existing theoretical analyses of hot-electron phenomena in sub-micron GaAs devices, largely simplified models have been adopted to make the calculation possible. [See (i) A. Majumdar et al., *Effect of gate voltage on hot-electron and hot phonon interaction and transport in a submicrometer transistor*, *J. Appl. Phys.* **77**, 6686 (1995), (ii) J. Schlee et al., *Phonon black-body radiation limit for heat dissipation in electronics*, *Nature Materials* **14**, 187 (2015) and (iii) J.Mateos et al., *Monte Carlo modelling of noise in advanced III–V HEMTs*, *J. Computational Electronics.* **14**, 72 (2015).] None of them would yield useful results if applied to the present work.

Below is the detailed sequence of our theoretical analysis, which we believe the only viable and reliable approach for the present study. Analytic expressions of $\langle T_e(x) \rangle$ and $P_{LO}(x)$ are given in Eqs. (3) & (4) in terms of the position dependent variables $n_\Gamma(x)$, $n_X(x)$, $T_\Gamma(x)$, $T_X(x)$ and $T_{LO}(x)$. For the evaluation of $\langle T_e(x) \rangle$ and $P_{LO}(x)$, we begin with the E -dependent quantities, $n_\Gamma(E)$, $n_X(E)$, $T_\Gamma(E)$ and $T_X(E)$, obtained from the Monte Carlo simulation on the hot electron transport in GaAs as the basis.⁴⁸ (In Ref. 48, the electric field E is assumed to be *uniform* so that the calculation is *position independent*. The computational domain needed is thereby dramatically saved, and reliable calculation is made possible.) The E -dependent quantities, $n_\Gamma(E)$, $n_X(E)$, $T_\Gamma(E)$ and $T_X(E)$, are converted to the position-dependent ones, $n_\Gamma(x)$, $n_X(x)$, $T_\Gamma(x)$ and $T_X(x)$, by using the profile of electric field, $E(x)$, derived for our device (shown in the figure above, Fig. S6). The position-dependent quantities are, in turn, modified further by noting the

known short-channel effects^{36,37} (in the section “VIII. Theoretical estimation of T_e and P_{LO} in two-carrier transport” in the revised Supplementary Information). The obtained quantities $n_r(x)$, $n_x(x)$, $T_r(x)$ and $T_x(x)$ (Fig. S8 a & b) are used to derive the profile of $\langle T_e(x) \rangle$ (Fig. S8b), which proves to well reproduce the experimental result (Fig. 3c). The profile of $T_{LO}(x)$ is derived from the fact that $T_{LO}(x)$ is close to but slightly lower than $\langle T_e(x) \rangle$ (see the new section “VI. Self-consistent determination of T_{LO} in a simplified model” in the revised version of Supplementary Information, and Fig. S8c). Finally, $P_{LO}(x)$ is derived (Fig. 3d and Fig. S6d), converted to $T_L(x)$ considering a broadening parameter, and shown to reproduce well the experimental profile of T_L (Fig. 3d).

-On its present form the modeling is too empirical to make the conclusions clear enough and unquestionable.

We are convinced that, in the revised version of the manuscript, our interpretation and conclusions are convincing and clear enough, being independent of particular models. We hope that Reviewer is satisfied with our new version of the manuscript, along with our response in the above.

Authors' response to Reviewer #2

The manuscript employs two complementary scanning probe techniques to probe energy dissipation in a III-V semiconductor device. In their first approach the authors probe evanescent fields by scattering radiation from a tip into a sensitive detector at a well chosen frequency and relate the intensity and spatial variation of the scattered radiation to the temperature of hot carriers. In a second experiment performed using a commercial scanning thermal probe in air, the authors attempt to measure the lattice (i.e. phonon) temperature. Finally from the measured spatial maps the authors argue that they have identified a novel mechanism that reduces energy dissipation from electrons. Overall, while the work is interesting, I find that there are several assumptions in the work that make the arguments somewhat speculative and the final conclusions somewhat weak. Please find below my comments:

Reply: We thank Reviewer for the careful reading of our manuscripts and his/her interest in our work. We are grateful to all of the comments, which are precious to us for they helped us a lot to distinctly improve the clarity of our discussion in the revised version of the manuscripts. Below we describe our response in detail to each point of Reviewer report.

1) Key to the interpretation of the results is the assumption that equations S1 and S2 are applicable even when the electrons are under highly non-equilibrium conditions. This is hard to justify.

Reply: We appreciate the comment of Reviewer for reminding us of the importance of giving discussion to justify equations S1 and S2. Our description in the original version of manuscripts might have been insufficient, having led to some misunderstanding of Reviewer. Equations S1 and S2 are theoretically justified and experimentally validated relations, applicable to the highly non-equilibrium electron system in the present study. This has been thoroughly discussed and experimentally confirmed in existing literatures [see e.g., Eqs. (3) and (6) in Ref. 30]. Nevertheless, our description in the original version of Supplementary Information may have been insufficient. In the new version, we have improved the description in *section "I. What is measured with SNoiM"* so that we hope the discussion is convincing enough by itself. We describe all the points in detail below, whereas some points may overlap with the earlier discussion in existing literatures.

First, we briefly summarize the essence. No rigorous theory exists to treat non-equilibrium systems in principle. Hence, the key issue is how to approximate the systems. Remarkable approximation applied in equation S2 is to characterize the highly non-equilibrium electrons by just one parameter T_e . This is justified by the fact that the energy exchange within the electron system is frequent enough to approximately establish T_e . This is assured in the present study because the electron-electron scattering rate, $1/\tau_{ee} \sim 1/(40\text{fs})$, is much higher than the electron-phonon energy relaxation rate, $1/\tau_{e-ph} \sim 1/(1.2\text{ps})$ in all the conditions. The justification here does not necessarily imply that the Fermi function in the thermal equilibrium at $T = T_e$ resembles the true electron

distribution function. Instead, the key is that, despite possible dissimilarities, any physical quantities obtained by assuming T_e give reasonable approximation to the true values. (This is a general theorem of the hot electron physics established in different theoretical frameworks including the Boltzmann transport equation. Particularly, equation S2 stands for the so-called "*hot-electron Shot noise*".) We wish to mention in addition that several review articles have cited our work of SNoiM. Among them, the recent theoretical work of Greffet and his collaborators [Opt. Express **29**, 425 (2021)] particularly mentioned to our two-temperature model (equation S2) and noted it to be an approximation. We stress that equation S2 is certainly an approximation but is practically the only realistic, reasonable and reliable approximation applicable to the present study.

We now proceed to detailed description. Equation S1,

$$u(z, \omega, T_s) = \rho(z, \omega) [\hbar\omega / \{\exp(\hbar\omega/k_B T_s) - 1\}],$$

is a rigorous relation theoretically derived in the thermal equilibrium condition (Refs.S8 & S9). Equation S1 is important here because it serves as the basis for deriving equation S2. Joulain and his collaborators [Sec. IV of *Phys. Rev. B* **68**, 245405 (2003)] showed theoretically that u given by equation S1 is detected in the far fields when thermally excited evanescent fields are scattered with a tip (i.e., by SNoiM) in the thermal equilibrium condition at T_s . Experimentally, the signals obtained by SNoiM on metals and dielectrics in the thermal equilibrium condition were confirmed to be given by S1. That is, SNoiM signals proved to be a product of a material-specific component ρ and a temperature-dependent component, and the former was quantitatively explained by the theoretically derived material-specific electromagnetic local density of states (EM-DOS) given in Refs. S8 & S9 (the left figure below), and the latter was confirmed to be proportional to $1/\{\exp(\hbar\omega/k_B T_s) - 1\}$ (the right figure below). Equation S1 was shown to quantitatively explain the

Material-specific component ρ :
Studied at fixed T (300K).

Data taken from Fig.6 in K.T.Lin, *et al.*, Review of Scientific Instruments **88**, 013706 (2017).

Temperature-dependent component :
Studied at a fixed point on given materials.

material-specific and the temperature-dependent characteristics of the signals of SNoiM studied on a variety of metals and dielectrics in the thermal equilibrium

condition.

Equation S2 is obtained by generalizing equation S1. In general, evanescent fields consist of the contributions from the random motion of conduction electrons and from the random lattice vibrations, so that the first term, ρ , in equation S1 is the sum of the respective contributions; viz., $\rho = \rho_e + \rho_L$. Equation S1 is hence rewritten as

$$u = (\rho_e + \rho_L)[\hbar\omega\{\exp(\hbar\omega/k_B T_s) - 1\}]. \quad (\text{S1})$$

In the condition when the two systems are out of equilibrium, the two contributions have to be distinguished by characterizing them with the respective effective temperatures, T_e and T_L , yielding equation S2 [eq. (6) in Ref.30],

$$u = \rho_e [\hbar\omega\{\exp(\hbar\omega/k_B T_e) - 1\}] + \rho_L [\hbar\omega\{\exp(\hbar\omega/k_B T_L) - 1\}], \quad (\text{S2})$$

As mentioned in the beginning of this section, the remarkable approximation implicit in equation S2 is to characterize the highly non-equilibrium electrons by just one parameter T_e . This approximation is justified by $1/\tau_{ee} \gg 1/\tau_{e-ph}$ in the present study. [Strictly, characterizing the lattice system by just one parameter, T_L , is not safely justified, but the discussion in this work is unaffected because the contribution from the lattice system is negligibly small ($\rho_L \ll \rho_e$) in our SNoiM as experimentally confirmed in this work.] The two-temperature model (equation S2) in the above was first introduced in Ref. 23 to successfully quantify T_e in the measurements of SNoiM. The validity of this analysis has been confirmed further in subsequent experiments [Weng, Q. et al., *Nano Lett.* **18**, 4220 (2018), and *Appl. Phys. Lett.* **114**, 153101 (2019)].

2) The SThM imaging is done in ambient conditions. Under such conditions conduction through air is expected to dramatically impact both the ability to do quantitative thermal imaging and the resolution of the measurements. Therefore, the SThM data is not very robust. In fact, as mentioned in the methods section, the height of the mesa structure on these devices is 100 nm. One would therefore expect significant temperature artifacts at the edges of the channel in SThM images, if the heat transfer is dominated by tip sample contact (see F. Menges, Nature Comm. (2016)). However, T_L does not show any artifacts and looks very continuous suggesting that there is significant averaging of temperature fields, possibly due to conduction through air.

Reply: We understand the Reviewer's concern about possible influence of the air-mediated thermal conduction. However, we are convinced that the influence is small and our measurements are reliable in both the quantitative terms and the resolution. Nevertheless we appreciate the comment of Reviewer for it reminds us of the importance of assuring the reliability of our SThM measurements. We have added brief notes in "section II. Scanning thermal microscope (SThM)" of the revised Supplementary Information.

Before responding to particular points, we wish to mention that an excellent spatial resolution (~ 50 nm) has been reported in the studies of SThM made in the ambient

condition [Luo *et al.*, *J. Vac. Sci. Technol. B*, **15**, 349 (1997), Gomes *et al.*, *Meas. Sci. Technol.*, **10**, 805 (1999), Shi *et al.*, *Appl. Phys. Lett.*, **77**, 4295 (2000), Shi *et al.*, *IEEE Proc. Int. Reliab. Phys. Symp.*, **38**, 394 (2000)]. The result was explained by a theoretical analysis showing that the air conduction is insignificant but the tip-sample contact is enhanced to dominate through bridging via liquid [see, e.g., Shi *et al.*, *J. Heat Transfer* **124**, 329 (2002)]. As described below, our experience is consistent with the results of those earlier studies made in the ambient condition.

As Reviewer notes, edge-related artifact signals are not distinctly visible in the SThM images shown in Fig. 1b. However, edge-related artifact signals are clearly visible in different measurement runs for the same device as displayed in the figure below. We found that the artifacts depend crucially on the probe tip used, the surface condition, the fine location of the scanned trajectory and the mesa-etch depth,

Edge artifacts in the SThM signal on the device of Figs 1 and 3 at $V_b = 8.0V$. Scan area is $10\mu m \times 10\mu m$.

suggesting that they are caused by irregular touches of the probe-tip side face to the mesa edge during the scan. When taking the data for presentation as shown in Fig. 1b and Fig. S4 (Supplementary Information, replotted below), we have carefully chosen the probe tip and the fine location for the scan trajectory to avoid the edge artifacts.

Fig. S4. SThM signal on the device of Figs 1 and 3 at $V_b = 8.0V$. Scan area is $10\mu m \times 10\mu m$. (a) 2D image. (b) 1D-scan. (c) Topographic image.

(Still a closer look at the data curves reveals that the signal slightly spikes at the edges.) Compared to the edge-related artifacts in the work of Menges *et al.*, *Nature Comm.* (2016), they appear to be less remarkable in our experiments. Supposedly, this is because (i) involved temperature rise of the target in our measurements is substantially

smaller ($\Delta T_L < 1$ K) and (ii) our probe tip is kept at room temperature T_R whereas it is heated up to 20-200K above T_R in the work of *Menges et al.*. In any case the presence of edge-related artifacts in our experiments proves the dominant role of the direct tip-sample contact and a minor role of the air-mediated thermal conduction.

A direct evidence of an excellent spatial resolution as well as the quantitative reliability of our SThM measurements is shown in the figure below, which displays results of additional experiments made in the ambient condition on a Joule-heated narrow metal wire (20 nm-thick NiCr) deposited on a SiO₂/Si substrate (a). (The measurements are similar to our earlier study of SNoiM; *Q. Weng et al., Near-Field Radiative Nanothermal Imaging of Nonuniform Joule Heating in Narrow Metal Wires, Nano Lett. 18, 4220 (2019)*). In the measurements on metals, the resolution and the quantitative accuracy can be directly confirmed because (i) the temperature changes sharply at the edge of the heated metal wire and (ii) the conduction electrons are in quasi equilibrium with the lattice so that quantitative comparison with simulation calculation is possible ((b),(e) ; *L. Yang et al., Simulation of temperature profile for the electron and the lattice systems in laterally structured layered conductors, EPL 128, 7001 (2019)*).

A hot-spot (b, c, d), showing up along the inner edge of the bended corner of the wire (currents $I = 1 \sim 8$ mA) due to local Joule heating caused by the current crowding effect, provides a convenient target for the test. A spatial resolution better than 100 nm is demonstrated by the sharp step-wise increase of the SThM signal at the inner-edge of the wire (d). (Edge related artifact signal is also seen at the outer edge of the wire.) The 2D-images of simulation (b), SNoiM (c; T_e), and SThM (d; T_L) are similar to one another. Especially, (e) shows that the amplitude ΔT_L derived from SThM is substantially accounted for by the theoretically expected values given by the simulation calculation, confirming the validity of our SThM measurements in absolute terms. (A discrepancy of a factor ~ 2.5 is present between the experiment and the simulation, which we interpret as a consequence of the tip-sample thermal-contact resistance R_{ts} noted by e.g. *F. Menges et al. 2016*.) In any case the discussion in the present work is unaffected by that amount of possible discrepancy.

3) For the data shown on Fig. 1 the images of T_e and T_L need to have a larger field of view. An increased field of view for T_L is crucial, since T_L could be just showing local cooling at the inlet of the channel due to Peltier or Thomson effect.

Reply: We greatly appreciate the comment of Reviewer on this important issue. In principle, Thomson effect may not be visible in our experiments because temperature gradient is not externally given, but Peltier effect can theoretically occur as Reviewer suggests because the heat flow is hindered at the constriction. However, the relevance of Peltier effect is strictly ruled out as discussed below. Nevertheless, noting the importance of this issue, we added discussion on the possibility of Peltier effect in the revised version of the manuscripts (main text, the last paragraph of section "Hot LO-phonon bottleneck in the electron transport").

Images of T_L in larger areas have been shown in Fig. S3 ($10\ \mu\text{m} \times 3\ \mu\text{m}$) and Fig. S4 ($10\ \mu\text{m} \times 10\ \mu\text{m}$) of Supplementary Information. As shown in the figure below (replot of Fig. S3 b & c), the T_L -distribution shows no place of local cooling, exhibiting smooth heating everywhere. In a still larger area ($120\ \mu\text{m} \times 120\ \mu\text{m}$), the temperature distribution is uniform without exhibiting any structures, as shown in the figure on the third page below, where conventional thermal microscope images at the same wavelength ($\lambda=14.5\ \mu\text{m}$) are displayed. As to the image of T_e in a larger (lead) region, it is completely certain that there should be no structures. Thus there is no experimental signature suggesting Peltier or Thomson effect.

Figure S3 b & c: Distributions of T_L obtained with SThM.

Peltier effect due to hindrance of heat flow at constrictions has been demonstrated in graphene constrictions [Achim Harzheim et al., Nano Lett. **18**, 7719 (2018), Xudong Hu et al., Small **16**, 1907170 (2020)]. In our experiments, however, Peltier effect is ruled out because the local heating and the local cooling would take place, respectively, at the channel entrance and the channel exit, **which is opposite to the observation** in

the present experiment. Theoretically, Peltier effect should not be visible in our experiments because its expected amplitude is too small as discussed below. Bulk Seebeck coefficient in the wide (lead) region of our n-GaAs device ($n=3.3 \times 10^{18}/\text{cm}^3$) is nearly $S_{bulk} \approx -100 \mu\text{V}/\text{K}$ at $T=300 \text{ K}$ [G.Homm et al., Appl. Phys. Lett. **93**, 042107 (2008)]. It follows that a heat flow $\dot{Q} \approx S_{bulk}TI \approx 7.5 \times 10^{-6} \text{ W}$ accompanies the current $I \approx 0.25 \text{ mA}$ for $V_b=8.0 \text{ V}$ (Fig.1) in the lead region. The absolute upper limit of Peltier effect is evaluated by assuming that the Seebeck coefficient reduces to zero, $S_{channel}=0$, at the constriction. Hence the upper limit of Peltier effect is $\pm \dot{Q}_{Peltier} \approx 7.5 \times 10^{-6} \text{ W}$, which is less than 1 % of the electrical input power (Joule heating), $P_{Joule} = V_{channel}I \approx 8.6 \times 10^{-4} \text{ W}$, in the constriction channel for $V_b=8.0 \text{ V}$, where $V_{channel}=3.45 \text{ V}$ is the net voltage drop along the constriction channel. Thus, Joule heating (or the effect of the electric field) completely dominates the kinetics of heat in our experiments, supporting the interpretation of our work. The relative amplitude of the Peltier effect with respect to Joule heating effect is roughly given by $\dot{Q}_{Peltier}/P_{Joule} \approx S_{bulk}T/V_{channel} \propto S_{bulk}/\rho_{channel}$. While the amplitude of bulk Seebeck coefficient S_{bulk} is similar between the two material systems, the distinct difference is the resistivity $\rho_{channel}$, which is by more than two orders of magnitude higher in the present GaAs constriction than in graphene constriction. Besides, Thomson effect, typically only a small correction, is also safely ruled out in our work because the (lattice) temperature gradient is very small and the Seebeck coefficient of n-GaAs does not have strong temperature dependence.

We wish to finally mention that our work focuses on the difference between the T_L - and the T_e -distributions. The difference itself is of significant physical implication that can never be attributed to Peltier effect. In several existing studies of nanoscale T_L -distribution,^{19,43,44} analysis is made by dividing T_L -profile into symmetric and antisymmetric parts with respect to the bias current polarity, and the former and the latter are interpreted, respectively as due to the Joule heating effect and the Peltier effect. It should be noted that this is an assumption not physically justified. We have demonstrated in the present experiments that, in the hot electron condition, the antisymmetric part of the T_L -profile can be entirely dominated by the electric-field-induced Joule-heat effect. This was made possible by the measurements of both T_e and T_L .

4)The estimated $T_L=300.8 \text{ K}$ does not seem to be consistent with a simple joule heating and thermal conductance calculation. Considering a substrate of GaAs with thickness of $250 \mu\text{m}$ and the shown geometry, and neglecting all the thermal contact resistances, a thermal resistance of $2.2 \times 10^4 \text{ K/W}$ can be obtained. This leads to a temperature rise of 44 K due to joule heating of 2 mW , which is much larger than that reported by the authors.

Reply: The estimation given by Reviewer does not appear to be appropriate. In real geometry Joule heat is generated in a limited area $\sim (1 \mu\text{m})^2$ of the device. Since the device is epitaxially grown on the GaAs single crystal substrate, the generated heat, in turn, spreads quasi-isotropically in 2π solid angle in the substrate without being hindered at the heterostructure interfaces as shown below. The heat flow thereby dilutes as $1/r^2$. The temperature rise at distance r from the hot spot is whereby

derived to be $\Delta T_L = P/2\pi r K_L$ with P the Joule heating power and K the lattice thermal conductivity of GaAs. Hence, we obtain $\Delta T_L \sim 0.6$ K at $r = 10$ μm for $P \sim 2$ mW and $K_L \sim 55$ W/m·K (GaAs), which is consistent with $T_L = 300.8$ K. (The overall heating of the GaAs substrate is far smaller because the total volume is much larger.) Our estimate in the above is also consistent with a more involved simulation calculation (L. Yang et al., EPL **128**, 7001 (2019)).

Aside from the estimate in the above, $\Delta T_L = 44$ K is strictly ruled out experimentally. Far-field thermography images (similar to the one shown in Fig. 2 of Weng et al., Nano Lett. **18**, 4220(2018)) are displayed in the figure below, which shows that the lattice temperature is not elevated discernibly ($\Delta T_L < 2$ K; *Otherwise it should be observed in the thermography image*) in a 120 $\mu\text{m} \times 120$ μm area by application of $P = 3.2$ mW to a similar GaAs device with a constricted channel.

5) Finally, the mechanism proposed by the authors does not make much sense to me. The authors state that heat dissipation occurs at the channel exit as indicated by the SThM signal. This means that the electrons collectively lose energy at the channel exit. Yet the prominent peak of the T_e occurs at this location, why is this? Further, when optical phonons are emitted and reabsorbed one would expect that the momentum of the electrons will be randomized and lead to a substantial electrical resistance, which would imply a local drop in voltage, which is undesirable and not an effect that is technologically useful as suggested by the authors.

Reply: We appreciate the critical comment of Reviewer. We have taken it seriously and supposed that our discussion in the original version of manuscripts may have been too compact for proper understanding. We have hence significantly improved the discussion in the revised version, so that it is longer but we believe the discussion is

much simpler to follow and transparent.

The first question of Reviewer, why the energy is primarily dissipated at the channel exit and why the prominent peak of T_e occurs at the channel exit, is an important issue discussed in our previous work (Ref. 23). Most simply described, the electrostatic potential energy ($-e\phi$) steeply falls along the constriction channel as depicted in the figure on the right. The electrons are accelerated by the electric field, $E = -\partial(-e\phi)/\partial x$, in the channel and drift along the channel without significant energy dissipation. Accordingly, the kinetic energy of electrons (or the effective electron temperature), $\langle \mathcal{E} \rangle = k_B T_e$, increases during the drift and reaches maximum at the channel exit. The highest T_e is thus established at the channel exit, and it gives rise to the most significant energy dissipation by emitting LO phonons there.

The second question of Reviewer about the possible degradation of device performance due to LO-phonon reabsorption process is also an important point discussed in earlier literatures (Refs. 1, 2, 41 and 42). Luckily, however, the degradation is insignificant in the electron system of this work because the additional frequent reabsorption of LO-phonons due to the hot-phonon bottleneck effect occurs in X valleys, where the conductivity of electrons is low and does not appreciably degrade the transport characteristics due to the occurrence of the effect. The degradation-free feature is directly confirmed in the current-voltage characteristics being not substantially affected by the onset of the hot-phonon bottleneck effect. This feature is commonly expected also in a number of modern electronic devices so that the effect discussed in this work will be of great general importance.

Aside from the particular questions of Reviewer in the above, he/she may require clearer explanation for the fundamental mechanism in general. Main issues of the present study are why energy dissipation is concluded to be absent throughout the channel and what is the mechanism. We believe that the discussions on these issues have been made distinctly clearer in the revised version. We do not repeat each discussion here but give a brief guideline below. (i) Explanation of why the energy dissipation is experimentally concluded to be absent is given from the 5th line from the bottom on page 6 through the 6th line from the top on page 7. (ii) In section “*Hot LO-phonon bottleneck in the electron transport*” (pages 7 ~11), discussion of the hot-phonon bottleneck effect is made explicit by defining T_{LO} with new Eq. (1) and introducing explicit judgement condition with new Relation (2). Figure 4 is newly added to help straightforward understanding of the effect. We have made compact the original section “*Semiquantitative analysis of electron transport with hot-phonon bottleneck*” in the revised section “*Hot-phonon bottleneck effect in the two-carrier transport*” (pages 11~14) in order to highlight the consequence of the valley transfer of

electrons and to elaborate the interpretation of the detailed fine structures in the profile of T_e and T_L . In Supplementary Information, we have newly added “*V. Estimation of E in the channel*” and “*VI. Self-consistent determination of T_{LO} in a simplified model*” to support easy understanding.

For all the reasons described above, I think the results described in the paper are not robust.

Reply: We hope to have thoroughly answered all the questions of Reviewer in this response. Together with the significant improvement made in the revised version of the manuscripts, we sincerely hope that he/she agrees that the experimental results as well as the interpretation of our work are robust enough.

Authors' response to Reviewer #3

Manuscript presents simultaneous experimental measurements of electronic and lattice temperatures in current-carrying GaAs/AlGaAs nanoelectronic devices. Authors argue that due to hot "LO phonon bottleneck" electrons pass through in quasiadiabatic manner (i.e. without net energy loss). Experimental results are compared with theoretical modeling of energy transfer between electronic and phonon subsystems. The analysis is based on thermal distribution and golden rule transfer rates assumptions. Ability to measure both temperatures simultaneously is really exciting. Thus, manuscript deserves to be published. There are several points authors may want to address before the publication:

Reply: We thank Reviewer for his/her careful reading of our manuscripts and for the affirmative comments on the scientific and technological significance of our work. In particular, we are grateful to Reviewer noting the uniqueness of our ability to measure both electronic and lattice temperatures simultaneously, which has never been realized. Thanks to valuable comments, we believe to have improved our manuscripts in the revised version. We hope that Reviewer will be satisfied by our detailed response to each point of the comments below.

1. On p.4 of the manuscript authors discuss "hot-phonon bottleneck effect", where emission of energy from electrons is compensated by absorption of energy from hot LO phonons. It seems, that the effect is common for any steady-state. Indeed in steady-state phonons are already hot enough so that net energy exchange with electron subsystem is zero. A short explanation would be helpful here.

Reply: We thank Reviewer for the question and the suggestion. The emission and the absorption of LO phonons cancel each other if the electrons and the LO phonons are in **equilibrium** ($T_e = T_{LO}$). This is, however, a far stricter condition than the **steady-state condition**. In our experiments, regardless of the amplitude of bias voltage, the energy fed from the electric field \mathbf{E} to the device flows in one direction as follows: $\mathbf{E} \Rightarrow \text{Electrons } (T_e) \Rightarrow \text{LO-phonons } (T_{LO}) \Rightarrow \text{Lattice/long-}\lambda \text{ acoustic phonons } (T_L) \Rightarrow \text{Environment } (T_R)$. The system is always in a **steady state** because the energy flows steadily, but the energy flow can occur because the system is in a **nonequilibrium state** given by $T_e > T_{LO} > T_L > T_R$. Both T_e and T_{LO} are elevated with increasing the bias voltage, but T_{LO} is more rapidly elevated than T_e in the electron system studied here (see new Fig. 4 in the revised version). It follows that the difference $T_e - T_{LO}$ gets smaller with increasing the bias voltage. In a range of sufficiently high bias voltages, $T_e \approx T_{LO}$ is realized so that the energy flow, $\text{Electrons } (T_e) \Rightarrow \text{LO-phonons } (T_{LO})$, is nearly blocked. This is the hot-phonon bottleneck regime. It should be mentioned that in this hot-phonon bottleneck regime, as well, the equal (but suppressed) energy flux is passing constantly through each step of the whole process; viz., $\mathbf{E} \Rightarrow \text{Electrons } (T_e) \Rightarrow \text{LO-phonons } (T_{LO}) \Rightarrow \text{Lattice/long-}\lambda \text{ acoustic phonons } (T_L) \Rightarrow \text{Environment } (T_R)$. So, the electron system is in a steady state but out of equilibrium whether or not it is in the hot-phonon bottleneck regime.

Following the suggestion of Reviewer, we have made the description distinctly clearer in the revised version of manuscripts. Especially, comprehensive discussion is added in section “*Hot LO-phonon bottleneck in the electron transport*”, where T_{LO} is defined with Eq. (1), and the condition for the occurrence of hot-phonon bottleneck effect is explicitly given by relation (2) and schematically explained with newly added Fig. 4. We believe that the clarity of discussion has been dramatically improved.

2. *Related to the point above: discussion of improbability of absorption of the LO phonons at low bias seems to be equivalent to saying that steady-state between electrons and LO phonons cannot be achieved at low bias. Only high bias leads to not energy exchange zero. It is not clear why this is the case: steady state seem to be achievable at any bias.*

Reply: We again appreciate the fundamental question, which reminds us that our description of the mechanism in the original version may have been insufficient. Our response to this question is essentially the same as the one given in the above. Briefly, *steady state* is achieved but *equilibrium state* is never reached at any bias condition. (Only at high bias condition in our experiments, seemingly-equilibrium-like state occurs between the electron and the LO-phonon systems with, $T_e \approx T_{LO}$. However, even in this condition, the electron system is largely away from the equilibrium with the LO phonons because Γ -valley electrons frequently are emitting LO phonons while X-valley electrons frequently are absorbing LO phonons.) At low bias, the steady state is established with a constant (but relatively small) energy injection by the electric field E and a constant energy dissipation through phonon relaxation dynamics (1 LO-phonon \rightarrow 2TO-phonon, namely Klemens channel), without any blocking effect. The phonon relaxation rate (~ 5 ps) is sufficient enough to keep to T_{LO} be the distinctly lower than T_e . At high bias, however, the rate of LO-phonon emission is so high that the relaxation rate is no longer sufficient (bottlenecked by the relative slow Klemens channel): LO-phonons are hence over-populated yielding a high effective temperature close to the electron temperature ($T_{LO} \approx T_e$). In this steady state, LO-phonons energy exchange frequently with hot electrons (~ 0.1 ps) with both emission (due to Γ -valley electrons) and absorption (due to X-valley electrons), leading to nearly vanishing net energy dissipation.

3. *On p.11: "SThM does not measure $T_{\{LO\}}$ nor $P_{\{LO\}}$..." Does it mean that if measurement of $T_{\{LO\}}$ was possible, one would get $T_{\{LO\}}=T_e$?*

Reply: Yes, $T_{LO} \approx T_e$ would be obtained if the effective temperature of LO phonons with small q -vectors (close to the Brillouin zone center) is measured. This should be indeed realized by the tip-enhanced Raman scattering experiments.

4. *On p.5 of Supplementary Information authors state: "the electron energy distribution,*

in either valley, is expected to deviate substantially from the Fermi function... Due to these approximations, the discussion in this work may be subject to slight quantitative modification." This statement sounds strange: all the analysis (experimental and theoretical) in the work is based on assumption thermal distributions for electron and phonon subsystems. If these assumptions do not hold, one in principle cannot introduce concept of temperature and then all the analysis fails.

Reply: We thank Reviewer for the kind and in-depth comment. The statements might have been misleading. In the revised version, we have changed the misleading statement *"the electron energy distribution, in either valley, is expected to deviate substantially from the Fermi function. - Due to these approximations, the discussion in this work may be subject to slight quantitative modification"* to *"the electron energy distribution function, in either valley, will not be accurately described by the Fermi function. - - Nevertheless, these are justified approximations, and the discussion and the interpretation in this work are valid"*.

5. *p.14 of Supplementary Information: "In these calculations of Ref.43... hot phonon distribution is not considered." Does it mean that possibility to reach steady-state between electron and phonon subsystems is removed from theoretical modeling? With mechanism of energy exchange between the two subsystems being the main point of the manuscript, removing it from theoretical simulations seems to be too drastic an approximation. A discussion would be very helpful here.*

Reply: Reference 48 (Požela & Reklaitis, *Solid-State Electron.*1980) considers phenomena in steady states but does not consider hot phonons because it assumes that the phonons are in thermal equilibrium with the environment; viz., $T_{LO} = T_L = T_R = 300$ K. This is evidently not a self-consistent assumption, but this is the most widely applied approximation. Actually it is difficult to analyze nonequilibrium phenomena by treating consistently all the relevant physical quantities; viz., two-types of electrons (T_Γ , T_X), LO-phonons (T_{LO}), and acoustic phonons (T_L). Even up to now there are still no reliable calculations available for the hot-phonon bottleneck effect. To avoid confusion, the statement *"In these calculations of Ref.48... hot phonon distribution is not considered"* has been corrected to *"In these calculations of Ref.48... phonons are assumed to be in thermal equilibrium with the environment (room temperature)"* in section *"VIII. Theoretical estimation of T_e and P_{LO} in two-carrier transport"* of the revised version of Supplementary Information.

REVIEWERS' COMMENTS

Reviewer #1 (Remarks to the Author):

In their revised version the authors have raised all doubts I had after reading the previous version. They have strongly improved the quality of their manuscript by replying clearly and in a detailed way to all reviewers' comments. I am convinced that this work is important and it will be of broad interest both for solid state physicists and engineers and I strongly recommend its publication in Nature Communication.

Reviewer #2 (Remarks to the Author):

The authors have made extensive changes to address my questions/concerns. I recommend acceptance and publication.

Reviewer #3 (Remarks to the Author):

Authors made a very good work in addressing questions raised in the first round of review. Changes made in the text resolve confusion explicitly indicating open character of phonon system (presence of environment), which immediately explains the observed difference in effective temperatures of electron and LO-phonon systems. The only worrying point is level of theoretical modeling presented in Section VIII of the Supplementary Information: it seems, that (at least at model level) more appropriate theoretical description (the one capable to describe phonon bottleneck) is relatively straightforward. Thus, it is not completely clear why authors have chosen to follow Ref.48, which seems to be too drastic an approximation. At the same time, this theoretical issue is of secondary importance, because the main (exciting) advancement is on experimental side. Thus, I recommend current version of the manuscript for publication.